# Anatomically aware simulation of patient-specific glioblastoma xenografts

**Adam A. Malik**[1]*, **Cecilia Krona**[2], **Soumi Kundu**[2], **Philip Gerlee**[1], **Sven Nelander**[2]

**1** Mathematical Sciences, Chalmers University of Technology and University of Gothenburg, Gothenburg, Sweden, **2** Department of Immunology, Genetics and Pathology, Uppsala University, Uppsala, Sweden

* maadam@chalmers.se

## Abstract

Patient-derived cells (PDC) mouse xenografts are increasingly important tools in glioblastoma (GBM) research, essential to investigate case-specific growth patterns and treatment responses. Despite the central role of xenograft models in the field, few good simulation models are available to probe the dynamics of tumor growth and to support therapy design. We therefore propose a new framework for the patient-specific simulation of GBM in the mouse brain. Unlike existing methods, our simulations leverage a high-resolution map of the mouse brain anatomy to yield patient-specific results that are in good agreement with experimental observations. To facilitate the fitting of our model to histological data, we use Approximate Bayesian Computation. Because our model uses few parameters, reflecting growth, invasion and niche dependencies, it is well suited for case comparisons and for probing treatment effects. We demonstrate how our model can be used to simulate different treatment by perturbing the different model parameters. We expect in silico replicates of mouse xenograft tumors can improve the assessment of therapeutic outcomes and boost the statistical power of preclinical GBM studies.

## Author summary

Glioblastoma is the most common and aggressive brain tumor in adults. To aid in research, it is common to use mouse models created by implanting patient-derived tumor cells, which provide a controlled system to study glioblastoma growth and treatment response. While these models are essential for preclinical studies, they are costly, time-consuming, and cannot easily capture the variability observed across patients. In this study, we introduce a simulation model that simulates how patient-derived glioblastoma tumors grow in the mouse brain. Our approach uses detailed maps of brain anatomy and a small number of biologically motivated rules to recreate how tumors expand, invade, and respond to their environment. By adjusting the model to match experimental data, we can reproduce case-specific

**Data availability statement:** Data and code available at https://github.com/maadam-123/mouse_xenograft_gbm_simulation.

**Funding:** The Swedish Research Council (2021-03224 VR) funded SN, Swedish Cancer Society funded SN, Knut and Alice Wallenberg Foundation (2022.057) funded SN, as well as the Swedish Foundation for Strategic Research (CCS23-0011) funded SN and PG. The funders had no role in study design, data collection and analysis, decision to publish, or preparation of the manuscript.

**Competing interests:** The authors have declared that no competing interests exist.

tumor growth and explore how different treatments might work. This "virtual laboratory" has the potential to reduce reliance on animal experiments, improve the interpretation of preclinical studies, and ultimately support the design of more effective therapies for glioblastoma.

## 1 Introduction

Glioblastoma (GBM), the most common primary malignant brain tumor in adults, poses a significant clinical challenge. Despite various treatment modalities, such as surgical intervention, radiation therapy, chemotherapy, or their combinations, patients with GBM face a grim prognosis, with a median survival of just over a year after diagnosis [1,2]. The primary obstacle in the surgical management of high-grade glioma lies in its infiltrative nature, as tumor cells have the capacity to migrate extensively through healthy brain tissue, infiltrating regions critical for patient survival [2].

In the late 1930s, Scherer's pioneering studies of GBM pathology delineated four patterns of invasion, encompassing subpial spread, accumulation near neurons or blood vessels, and migration along white matter tracts [3]. Today, we know that the dynamic interaction between GBM cells and different anatomical niches is crucial in mediating, facilitating, and shaping spread of the tumor. For instance, invasion along blood vessels presents notable advantages for glioma cells [4]; The perivascular space, characterized by its ample fluid content, offers minimal physical hindrance. Additionally, the basement membrane showcases distinctive extracellular matrix (ECM) proteins that facilitate the movement of glioma cells. Of note, invasion along preexisting microvessels in the perivascular space is likely a VEGF-independent mechanism of tumor vascularization , meaning that this process likely cannot be suppressed by VEGF inhibitors [5,6]. In addition to the perivascular niche, the interaction with neurons and nerve fibers is also a crucial mediator of tumor spread. Radiological data strongly support that GBMs follow white matter fibers and data from animal models support distinct invasion rates in gray matter. Jointly, the tumor cells' interactions with blood vessels and white matter cause a high degree of anisotropy of the tumor, and contributes to the difficulty in determining a region for surgical resection or radiotherapy [7,8]. Amounting evidence in preclinical models also suggests that distinct molecular mechanisms may be at play in mediating invasion along different routes [9–13]. Jointly, these observations motivate investigation into the quantitative details of GBM invasion. Towards this goal, it is important to develop a quantitative understanding of how diverse invasion outcomes arise in GBM.

This paper seeks to align two distinct approaches to study brain tumor invasion: mouse patient-derived cell (PDC) xenografts and patient-specific, agent-based simulation. Orthotopic xenografts from patient-derived primary cells replicate many aspects of human GBM, including the formation of a core lesion with increased mitotic index, pleiomorphic cells, and neoangiogenesis [14–16]. Also, in similarity with the original tumors, xenografts primarily spread along the vasculature, through white matter structures, or diffusively through the parenchyma [15,17,18]. The hypoxic

niche and the perivascular niche are also important in driving tumor growth and invasion [18–20]. Comparisons of ortho-topic xenografts derived from different source patients have shown case-specific survival times and growth patterns [15, 21], providing a possible way to elucidate mechanisms of GBM-brain interaction.

In recent years, mathematical and computational models have become increasingly prevalent in cancer research, par-ticularly in the context of understanding tumor growth. One common modeling approach for describing the anisotropic growth of gliomas involves using a continuum description of a population of cells subject to diffusion and proliferation. For example, the Proliferation-Invasion model [22], published in 2000, uses two different diffusion constants in white and gray matter. More sophisticated continuum models have since been developed, with some incorporating adhesion mechanisms between glioma cells and ECM components [23,24]. This approach to modeling has also been extended to incorporate advanced imaging data such as Diffusion Tensor Imaging to support the simulations and assess therapeutic effects [24].

Another group of mathematical models are those which consider individual cells as the entities of interest. These mod-els can be based on cellular automata and its extensions or random walks. For example, a lattice-gas cellular automa-ton was developed to model glioma cell invasion in the brain, incorporating diffusion tensor data [25]. Another group of agent-based models involves random walks, with some models using diffusion tensor data to guide cell migration [26,27]. A benefit of such agent-based models is that they are based on explicit assumptions regarding cell behavior, rather than phenomenological principles of macroscopic behavior. It is possible to derive macroscopic descriptions from microscopic descriptions in certain cases [23,28], but it is a non-trivial task and it has its own limitations. We selected an agent-based modeling approach for several reasons. First, it allows for the straightforward integration of multiple biologically realistic mechanisms and offers flexibility for future extensions. Second, we aimed to focus on cellular-level dynamics, rather than adopting a macroscopic, population-level perspective. However, agent-based models are typically more computationally intensive, which may become a limiting factor depending on the available computational resources.

While anisotropic spread induced by white matter invasion has been extensively studied, the influence of perivascu-lar invasion on glioma growth has not received the same level of attention. In [5], an agent-based model was developed and compared against experimental data obtained from mouse xenograft experiments, where cells were assumed to be attracted to blood vessels. Some studies have incorporated both white matter and vasculature. In [29,30], a system of par-tial differential equations (PDEs) was used to model a population of normoxic, hypoxic, and necrotic tumor cells, along with vasculature and angiogenic factors. Tumor cells were assumed to migrate more rapidly through white matter than gray matter; however, vasculature did not influence migration. In [31] a system of PDEs was used to model tumor growth and blood volume fraction, where the diffusion coefficient is derived from local mechanical stresses. To the best of our knowledge, no previous study has investigated white matter and perivascular invasion simultaneously.

In this work we fill this gap by developing a model that explicitly incorporates both white matter and blood vasculature. The model has four parameters which we fit to data: migration rate, proliferation rate, white matter preference and blood vessel preference. By fitting these model parameters to data from 8 different xenografts, we have the goal to quantify the different invasion phenotypes observed in experiments. In particular, we will aim to answer the following questions. (i) Does adding explicit blood vasculature into the model add anything in terms of predictive power? (ii) Can our model cap-ture the growth patterns seen in mice? We end by demonstrating how the model can be used to simulate different types of treatment scenarios.

## 2 Results

### 2.1 Anatomy-aware simulation of GBM xenografts

Histological sections from xenografted GBM typically present with a central tumor lesion, together with evidence of spread in the perivascular and white matter compartments (Fig 1A). Yet, relatively little is known about the relationship between such phenotypes and the underlying dynamic processes, such as the rate of cell proliferation and migration. To clarify this, we propose a simulation strategy based on three main components:

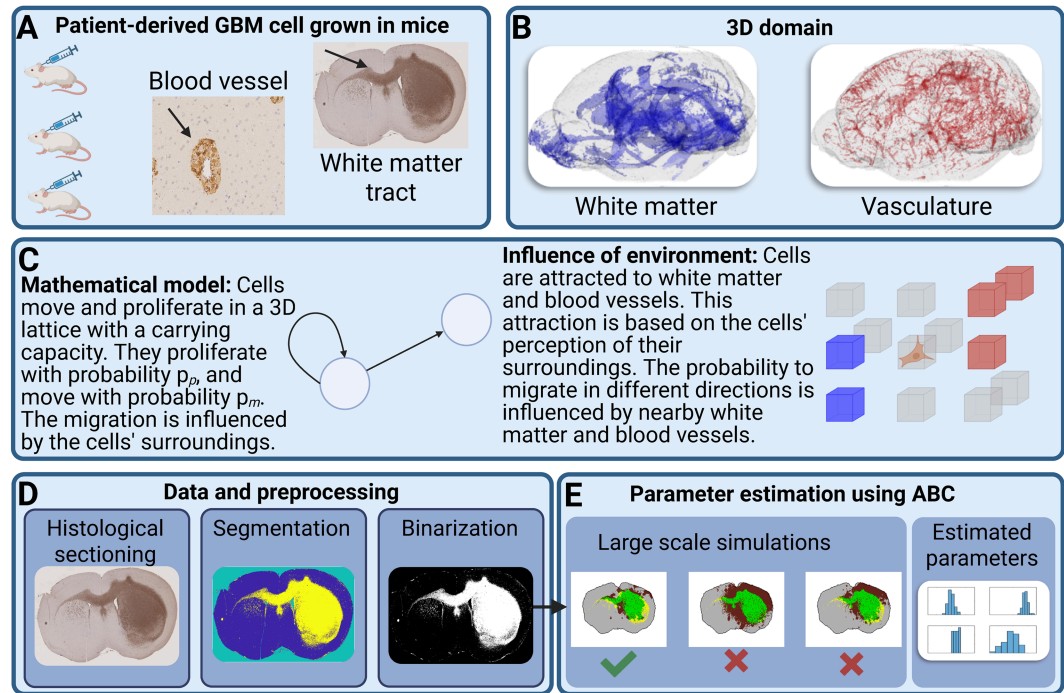

**Fig 1**. **Figure illustrating our workflow.** (A) PDCs have been grown in mice, followed by histological sectioning and staining. Perivascular growth and invasion along white matter tracts is present. (B) Our mathematical model incorporates 3D maps of vasculature and white matter tracts. (C) The mathematical model is an agent-based model where cell migration is influenced by white matter and vasculature. (D) Data is segmented before being used for model fitting (E) using Approximate Bayesian Computation. Created in BioRender. Nelander, S. (2025) https://BioRender.com/tj7y5g8.

First, the simulation is done on a 3D scaffold (map) that describes the normal mouse brain anatomy (Fig 1B). This was obtained by magnetic resonance Diffusion Tensor Imaging (DTI) data from normal adult C57BL/6J mouse brains with a resolution of 43 $\mu$m [32]. Using this data, we calculated white matter regions by thresholding the fractional anisotropy (FA) at 0.5. Intuitively, FA is a DTI-derived score that captures to what degree the water molecules in each small region of the brain are free to move, ranging from free movement in all directions (FA=0) to being constrained to a single direction (FA=1). We also incorporated brain vasculature data from a previous study that combined tissue-clearing microscopy with image processing to capture the vasculature down to the capillary level [33]. Both 3D structures can be seen in greater detail in Fig 2.

Second, we developed a mathematical model using an agent-based approach (Fig 1C). In this model, cells migrate and proliferate within a three-dimensional lattice, and changes occur at discrete time steps. The total number of cells is denoted by $N_t$, and each lattice site has a carrying capacity of $K$ cells. Cells have probabilities of proliferating ($p_p$) and migrating ($p_m$) at each time step, with their migration influenced by their microenvironment, particularly white matter and blood vessels. We introduced parameters, $w_{wm}$ and $w_{bv}$, representing the strength of attraction toward white matter and blood vessels, respectively. The probability of migrating in different directions depends on these parameters.

The third aspect of our methodology involves data preprocessing (Figs 1D and S1) and parameter estimation using Approximate Bayesian Computation [34] (ABC) (Fig 1E). The key idea of ABC is to run a simulation across multiple, randomly picked parameter values. By subsequently comparing the simulations to the experimental data (histological slides of mouse brains), a range of credible parameters is obtained. We used summary statistics based on the geometric properties of tumors to compare simulated tumor growth with real tumor data. These summary statistics include the number of connected components, area, perimeter, filled area, and eccentricity. We performed regression adjustment with a logit

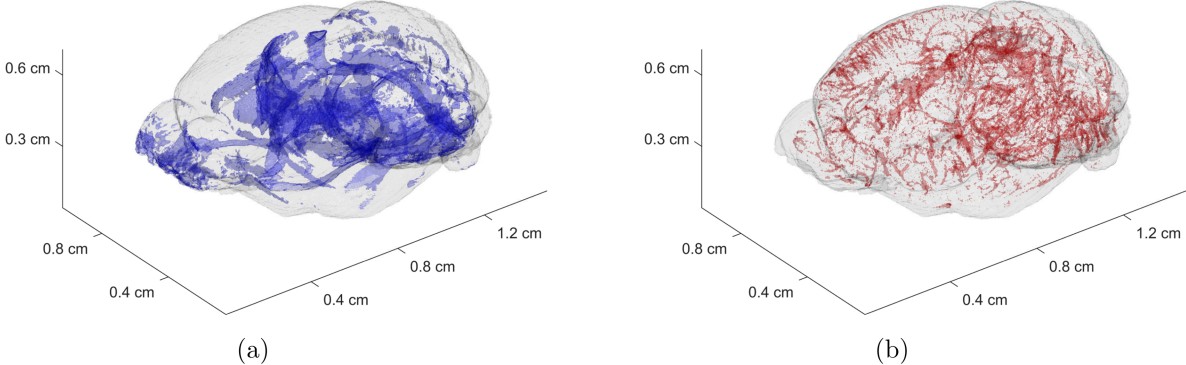

**Fig 2**. **Figure illustrating the anatomical structures in the mouse brain.** (a) 3D map of white matter. Derived from previously published diffusion tensor imaging dataset. (b) 3D map of blood vasculature. Derived from previously published dataset obtained by multi-dye staining and DISCO tissue clearing.

transform to ensure the adjusted posterior samples were within the same range as the respective prior distributions [35]. The parameters include $p_m$, $p_p$, $w_{wm}$, and $w_{bv}$, defined in Methods. Throughout this paper, we performed 5000 simulations per model and retained the best 50 parameter sets (top 1%). While a larger number of simulations would likely have improved the results, we limited ourselves to 5000 runs per model due to the computational demands of simulating such large-scale models.

Overall, our approach involves data preprocessing, mathematical modeling, and parameter estimation to develop a comprehensive model for glioblastoma growth. This model incorporates both anatomical influences and agent-based simulation for a more accurate representation of tumor behavior.

## 2.2 Improved estimation of parameters by geometric similarity measure

The task of comparing the location, size and shape of different tumors, obtained from histological sections or from simulations, is a challenging endeavour. In order to make a quantitative comparison between tumors, a metric has to be chosen. A frequently used metric is the Jaccard index, which measures fraction of overlap between images. Different metrics have their own strengths and weaknesses. For this reason we have developed our own metric that is based on geometric similarity, rather than spatial overlap. The goal is to investigate if histological sections together with our geometric similarity metric is sufficient to estimate the four growth and invasion parameters ($p_m$, $p_p$, $w_{wm}$, and $w_{bv}$) of our simulation model.

To investigate this idea, we first generated synthetic datasets by varying each parameter within specified ranges and conducting simulations (see Methods for details). By testing two (low vs high) values for each of the four parameters, we obtained $2^4 = 16$ simulations from which we extracted virtual histology slices, with clear differences in tumor size and invasion phenotype (Fig 3). Given such "ground truth" data, we evaluated how well the ABC algorithm performed in three different cases. Firstly, we use rejection-ABC with the Jaccard index as a summary statistic. Secondly, we use rejection-ABC with our own geometric summary statistic. Thirdly, we use rejection-ABC with our geometric summary statistic along with regression adjustment (using an Epanechnikov kernel as explained in Sect 3.4).

For each of the 16 datasets, we calculated the relative error between the true parameters $p_m, p_p, w_{wm}, w_{bv}$ and the estimated parameters for the 50 accepted parameters using the formula given in (4). The overall error for a specific method is then the average error over the 50 accepted parameters, as defined in (5). The averaged error $E$ over the 16 datasets was 0.857 for the Jaccard measure, 0.856 for the geometric measure without regression adjustment, and 0.512 when using the geometric measure with regression adjustment. The errors for the 16 cases are visualized in Fig 3. Based

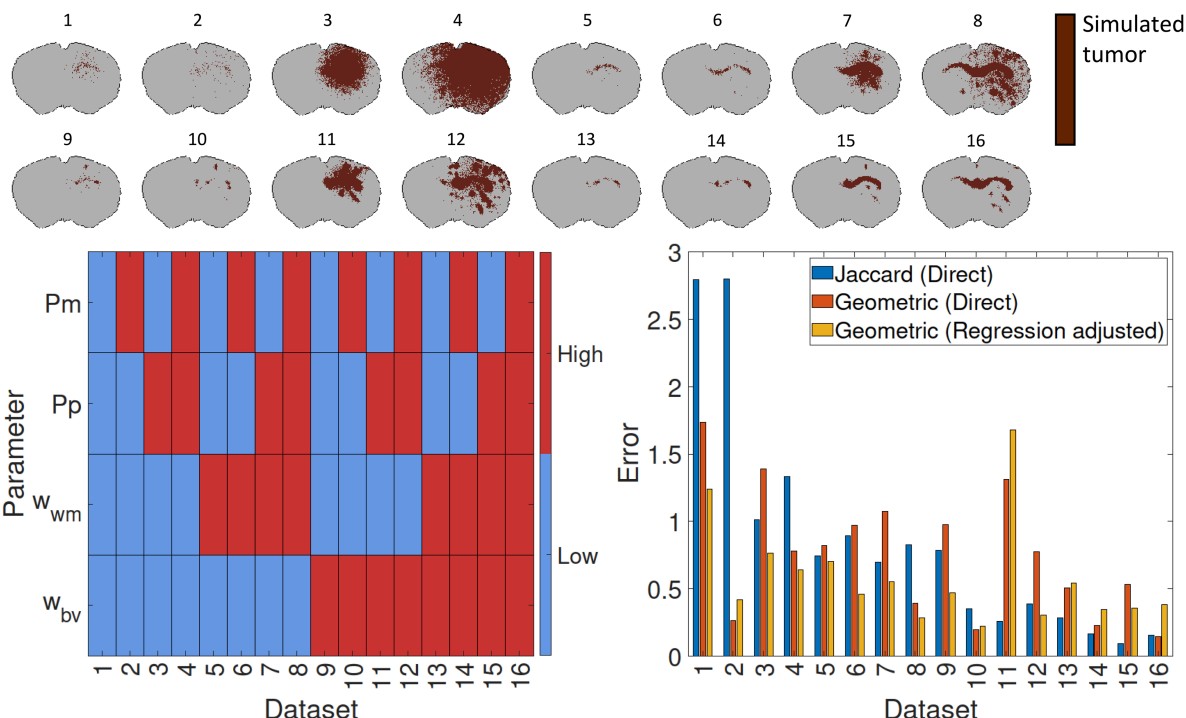

**Fig 3. Illustration of the synthetic testing of the modelling framework.** The 16 distinct synthetic tumors are shown at the top. The 16 tumors are generated by alternating the four parameters between low and high values (bottom left). The resulting errors are shown in the histogram (bottom right).

on these results, we opted to use the geometric measure with regression adjustment to fit the model to the histological sections from the PDC xenografts.

## 2.3 Model simulations reproduces overall size and tumor morphology

We fitted the full model that contains both white matter and blood vasculature to a total of eight PDC xenograft experiments (four cell lines, 2 replicates each). The results are reported in Table 1. The numbers represent the point estimate followed by the 10% and 90% percentiles. We also present the largest value of $\delta$ (the acceptance threshold), defined in (3), that allowed for exactly 50 parameters to be accepted. This value gives some insights into how well the simulated tumors fits the real tumors for each case. A large value of $\delta$ means that in order to accept 50 parameters, the accepted error threshold for the summary statistics was large, whereas a smaller value of $\delta$ implies that the overall difference in

**Table 1.** Estimated parameter values of the eight experiments, along with the 10% and 90% percentiles in parentheses.

| Tumor number | Cell line | $P_m$ [μm/h] | $P_p$ [1/day] | $w_{wm}$ | $w_{bv}$ | $\delta$ |
|---|---|---|---|---|---|---|
| 1 | U3013MG | 9.12 (1.96, 22.82) | 0.32 (0.28, 0.37) | 0.31 (0.15, 0.49) | 0.69 (0.5, 0.87) | 0.98 |
| 2 | U3013MG | 20 (5.6, 41.5) | 0.19 (0.14, 0.24) | 0.38 (0.26, 0.51) | 0.62 (0.34, 0.8) | 0.58 |
| 3 | U3230MG | 6.67 (5.2, 7.95) | 0.22 (0.22, 0.23) | 0.49 (0.21, 0.73) | 0.51 (0.25, 0.7) | 0.9 |
| 4 | U3230MG | 3.9 (1.52, 5.1) | 0.23 (0.18, 0.26) | 0.64 (0.19, 0.89) | 0.12 (0.04, 0.24) | 0.47 |
| 5 | U3033MG | 4.28 (1.8, 7.2) | 0.09 (0.067, 0.13) | 0.15 (0.02, 0.46) | 0.27 (0.04, 0.49) | 0.38 |
| 6 | U3033MG | 1.87 (0.9, 2.7) | 0.12 (0.89, 0.16) | 0.15 (0.02, 0.28) | 0.32 (0.14, 0.54) | 0.42 |
| 7 | U3062MG | 1.72 (0.22, 4.4) | 0.035 (0.028, 0.04) | 0.39 (0.08, 0.65) | 0.24 (0.1, 0.53) | 0.53 |
| 8 | U3062MG | 4.5 (1.7, 8.6) | 0.019 (0.014, 0.025) | 0.16 (0.03, 0.3) | 0.34 (0.14, 0.52) | 0.7 |

summary statistics was smaller. Hence $\delta$ can be viewed as a measure of the inverse quality of the fit. We also present the best fit for each tumor in Fig 4.

We can see that the overall size, and whether the tumor is diffuse or has a distinct boundary is captured well by the model simulations. However, the exact spatial overlap does not always agree, which is apparent in e.g. Tumors 1, 7 and 8. This is an inherent feature of the metric used, which does not take spatial overlap into account. The best fit of tumor 2 shows a very small collection of cells, in agreement with the real tumor. However, it did not capture the few single cells that have migrated far away in the real tumor. The fits shows good agreement in terms of overall size, and in most cases also general shape. It is noticeable in tumors 3, 5 and 6 that the growth is limited at the top of the tumor edge. It appears

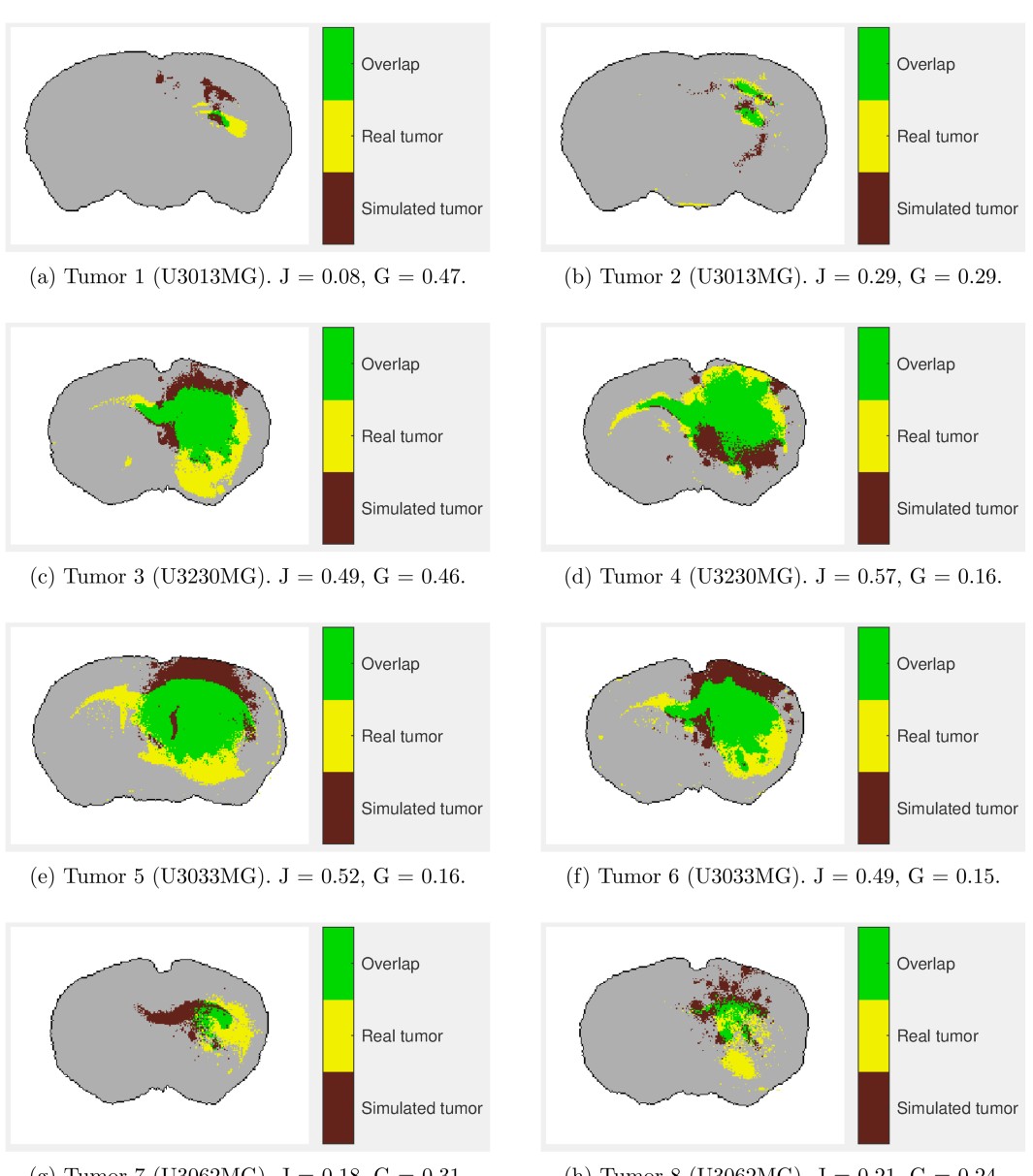

(a) Tumor 1 (U3013MG). J = 0.08, G = 0.47.

(b) Tumor 2 (U3013MG). J = 0.29, G = 0.29.

(c) Tumor 3 (U3230MG). J = 0.49, G = 0.46.

(d) Tumor 4 (U3230MG). J = 0.57, G = 0.16.

(e) Tumor 5 (U3033MG). J = 0.52, G = 0.16.

(f) Tumor 6 (U3033MG). J = 0.49, G = 0.15.

(g) Tumor 7 (U3062MG). J = 0.18, G = 0.31.

(h) Tumor 8 (U3062MG). J = 0.21, G = 0.24.

**Fig 4**. **Best fit for each of the eight tumors.** J indicates the Jaccard index (higher is better) and G geometric distance (lower is better).

to be another anatomical structure which is not white matter, which has influenced the growth. Tumors 3-7 exhibit growth along corpus callosum, which is replicated in the simulations in Tumors 3,4 and 6.

By inspecting the fitted parameter values in Table 1, we can see that there is some agreement between the replicates for each of the four cell lines. For U3013MG (Tumors 1,2), there is good agreement in terms of $w_{wm}$ and $w_{bv}$, but less agreement for $P_m$ and $P_p$. However, it is worth pointing out that tumor 2 appears to be spread over a larger area, and hence expected to have a larger migration rate $P_m$.

For U3230MG, (tumors 3 and 4) both replicates agree well for $P_p$ and $w_{wm}$ but not as well for $P_m$ and $w_{bv}$. The cause for this is not obvious. However we observe that their direction of spread appears similar along the corpus callosum, but their morpholoigies differ significantly apart from that.

The fitted parameters of U3033MG agree well for all parameters, except for $P_m$. However, tumor 5 (with larger estimated $P_m$) is visibly larger than tumor 6. Tumor 5 also grew for 165 days instead of 136 for tumor 6. It is therefore difficult to disentangle how much of the largest tumor is due to the slightly longer growth and how much is actually due to larger migration rates.

For U3062MG (tumors 7 and 8) there is some agreement. However, the spatial overlap between the simulated tumors and the real tumors is poor for both tumors. The real tumors also show somewhat distinct qualities, with tumor 7 appearing to be compact with a rather distinct boundary, whereas tumor 8 is more diffuse.

## 2.4 Addition of blood vasculature improves model fit

Since the combination of white matter and blood vasculature is a novel feature of our mathematical model, we tested if this new extension adds to the predictive capacity of the model. To evaluate this, we considered two alternative models, one full model (as above), and one white matter-only model (without vasculature). We performed 5,000 simulations for each model (using the same parameter ranges) and pooled all simulations into a set of 10,000. For each of the eight tumors, we went on to identify the 50 best parameter sets. To compare the merit of each model (full vs white-matter-only) we calculated the fraction of the 50 best simulations that came from either model. (The rationale for this comparison comes from the theory of ABC; more probable models will be over-represented, while less probable models will be under-represented or absent [35]). In six of the eight cases the full model had stronger support, and in the two cases where the white-matter only model was preferred, the fraction of accepted parameters was close to 50% from each of the two models (Table 2).

## 2.5 Simulating treatment response

In this section we conduct simulation experiments which are designed to resemble the outcome when treating cells with hypothetical drugs that can reduce migration or proliferation. The aim is to determine if there is an inter-tumor heterogeneity in the response to different combinations of drugs.

**Table 2**. Model support calculated as the fraction of accepted parameters from each of the two models.

| Tumor number | Cell line | Fraction of accepted parameters Full model | Fraction of accepted parameters White matter-only model |
|---|---|---|---|
| 1 | U3013MG | 0.66 | 0.34 |
| 2 | U3013MG | 0.48 | 0.52 |
| 3 | U3230MG | 0.58 | 0.42 |
| 4 | U3230MG | 0.84 | 0.16 |
| 5 | U3033MG | 0.72 | 0.28 |
| 6 | U3033MG | 0.6 | 0.4 |
| 7 | U3062MG | 0.82 | 0.18 |
| 8 | U3062MG | 0.46 | 0.54 |

To this end we perform the following type of experiment. We assume that there is a treatment that knocks out the mechanism (proliferation or migration) for a proportion of all cells. For example, we assume that each cell either stops migrating, or stops proliferating. The distribution of cells into these subpopulations varies between a 100% - 0% distribution (only anti migration drug), then a 90% - 10% distribution, until we reach the final distribution of 0% - 100% (only anti proliferation drug). We implement the drug action by assuming that, in each time step, each cell either does not migrate or does not proliferate, and the probabilities are given by the assumed drug distribution. For example, when we simulate a distribution corresponding to 70% anti migration and 30% anti proliferation treatment, the cell fails to move with probability 0.7, and if it did not fail to move, it fails to proliferate.

Our experimental design serves the purpose of incorporating a type of effect/side-effect trade-off. Provided that there is some limit to the allowed or tolerated amount of side effects, there may be cases where a tumor can be more efficiently treated by reducing the dose of one treatment (and hence the effect), and instead allowing the introduction of a secondary one. We use as a starting point the fitted parameters from tumors 1-8 and initiate a tumor at day 0 and let it grow for 30 days unhindered. At day 30 the treatment is introduced and remains for another 60 days. To compare the treatments, we compare the final tumors to the case of an untreated control tumor growing for 90 days.

To measure the outcome of the different drug combinations on the growth of tumors we use two different measures of tumor size. The first is the total cell count, and the second is the convex hull of the tumor. We chose to use the convex hull as a proxy for the spatial distribution of cells, which the cell count does not account for. We illustrate our results by comparing the treated tumor's cell count and convex hull relative to the untreated tumor's cell count and convex hull in Figs 5 and 6.

We can see that different tumors respond differently to the different drug combinations.

For tumors 1-6 it appears that the cell count is lowest when the tumor is treated with only anti migration or only anti proliferation drugs, with a maximum somewhere between the two extreme cases. For tumors 7 and 8 the cell count is lowest when treated with only anti proliferation drug. The convex hull on the other hand seem to be lowest when treated only with an anti migration drug, for all eight tumors, even tough the response differs between the eight tumors. This is expected to some extent, as the convex hull is capturing the spatial spread of the tumor. On the other hand, in the cases of tumors 1-4 and 6, we see that there exists an intermediate point where the convex hull starts to decrease again, with increasing doses of the anti proliferation drug. Interestingly, tumors 5,7 and 8 does not show the same qualitative behavior. It appears that the convex hull more or less increases monotonically with increasing proportions of anti proliferation drug, although tumors 5 and 7 seem to exhibit a small decrease in convex hull when going from a 10%-90% treatment to a 0%-100% treatment, in favor of anti proliferation drug.

## 3 Material and methods

### 3.1 Ethics statement

All mouse experiments have been pre-approved by the regional animal research ethical committee (permits C41/14, 5.8.18-02571-2017).

### 3.2 Data

Patient-derived cells (PDCs) obtained from the HGCC biobank [15,36] were labeled with a GFP-luciferase construct (pBMN(CMV-copGFP-Luc2-Puro, Addgene plasmid #80389, a kind gift from Prof. Magnus Essand, Uppsala University), to enable monitoring of tumor growth by in vivo bioluminescence imaging on a NightOWL imaging system (Berthold Technologies). Following puromycin selection, 100.000 cells were injected into 6-11 weeks old female NOD/MrkBomTac-Prkdcscid (Taconic), NOD.CB17-Prkdcscid/scid/Rj (Janvier Labs), NOD.Cg-Prkdcscid Il2rgtm1Sug/JicTac (Taconic), and Rj:NMRI-Foxn1 nu/nu (Janvier Labs) under 2-4% isoflurane anesthesia. The cells were injected into a precise location of the striatum using a stereotactic frame (KOPF model 940) and bregma as reference point 0: AP 0, ML 1.5 (right), DV -3.0.

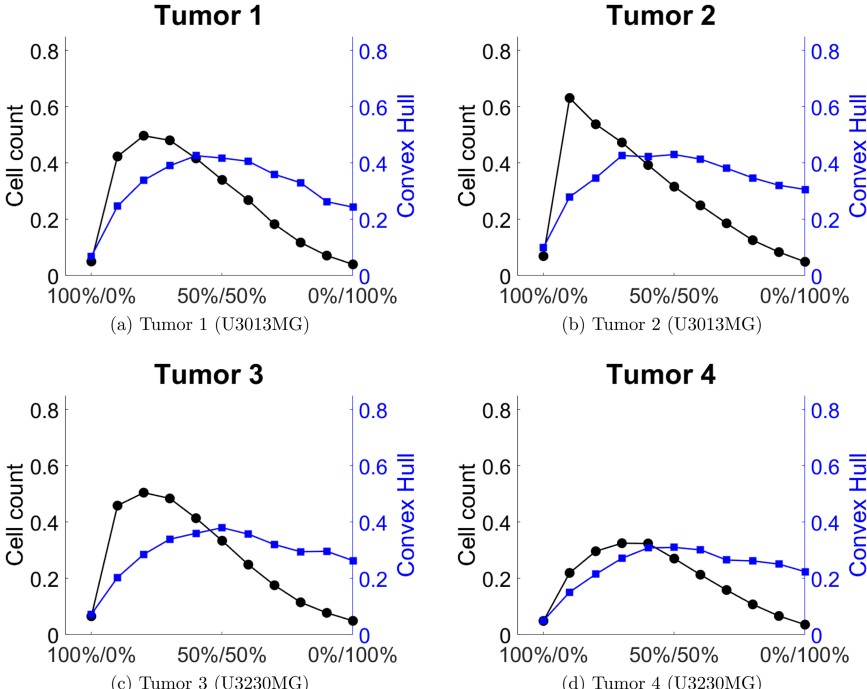

**Fig 5**. **Cell count and convex hull of the treated tumor relative to the untreated case for different combinations of anti migration and anti pro-liferation drugs for tumors 1-4.** The combination 100%/0% corresponds only anti migration drug, and 0%/100% corresponds to only anti proliferation drug.

Mice were given 5 mg/kg carprofen, (Orion Pharma Animal Health, Sweden) subcutaneously before injection and again the following day. All mice were monitored closely until the endpoint defined by the occurrence of the first of the following three events; (1) luciferase levels increase by a magnitude of 100-1000, (2) mice display neurological symptoms or weight loss exceeding 10% of their maximum weight, or (3) 40 weeks has passed since injection of cells.

At the experimental endpoint, mice were euthanized using a gradually increasing concentration of $CO_2$ in air, followed by cervical dislocation. Intact brains were harvested and saved in 4% buffered formaldehyde (Histolab products AB, Sweden) after brief washing in PBS. For paraffin embedding, each brain was first sliced coronally into five equal-thickness pieces (2 mm) and processed for dehydration and paraffin embedding using an automated tissue processor system (TPC15 DUO, Medite Medizintecknik, Germany). The tissue blocks were prepared using a consistent pattern alignment with the five slices to generate comparable 3-um thick sections after cutting and mounting on glass slides.

To visualize tumor cells, sections were stained with anti-human specific NuMA (ab 97585, Abcam) and Stem121 anti-bodies (Y40410, Takara) following standard procedures and visualized with DAB Quanto (Thermo Fisher Scientific) fol-lowing incubation with horseradish peroxidase-conjugated secondary antibodies; goat anti-mouse IgG (AP308P, Millipore) and goat anti-rabbit IgG (AP307P, Millipore). To generate high-resolution images, sections were scanned in a pyramidal structure with the highest magnification of 20x and resolution of 0.5 $\mu$m using an Aperio ScanScope XT Slide Scanner (Aperio Technologies, Vista, CA, USA) at the Swedish Science for Life Laboratory (SciLifeLab) Tissue Profiling facility at Uppsala University.

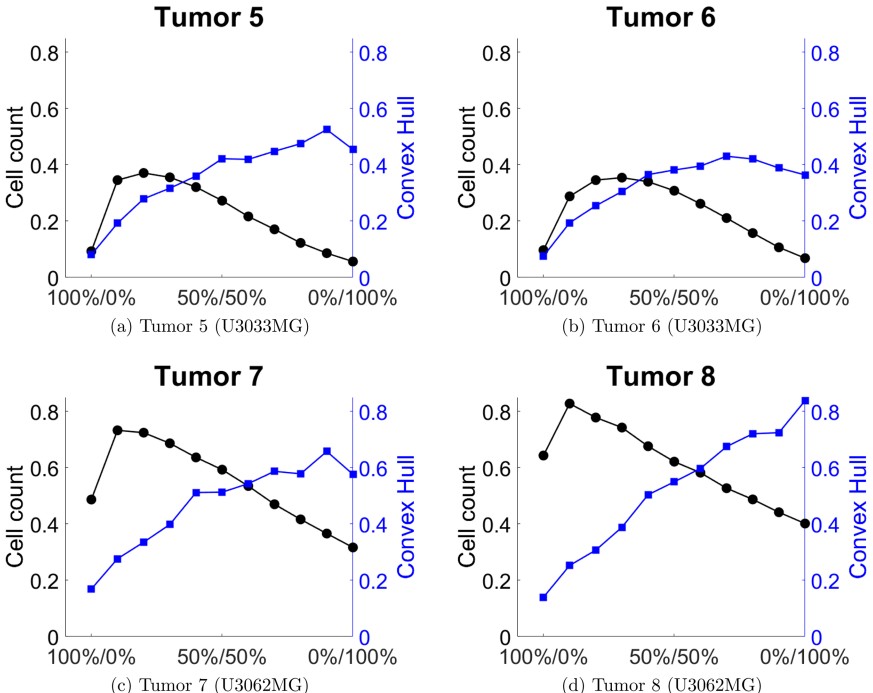

**Fig 6**. **Cell count and convex hull of the treated tumor relative to the untreated case for different combinations of anti migration and anti proliferation drugs for tumors 5-8.** The combination 100%/0% corresponds only anti migration drug, and 0%/100% corresponds to only anti proliferation drug.

The clinical information of the patient derived cell lines used in this study are summarized in Table 3.

## 3.3 Mathematical model

We consider an agent-based model where cells are migrating and proliferating in a three-dimensional lattice with spacing $\Delta x$, $\Delta y$ and $\Delta z$, and where changes to the population occur at discrete time steps. We denote by $N_t$ the total number of cells at time step $t = 0, 1, \ldots, T$ of duration $\Delta t$. Each lattice site has a carrying capacity and can accommodate at most $K$ cells at the same time. In each time step, $N_t$ agents are chosen independently at random, one at a time. When an agent is chosen it is given the opportunity to proliferate, with probability $p_p$, provided the number of cells is less than $K$ in the current voxel. The offspring is placed in the proliferating cell's voxel, but the offspring cannot proliferate or move in the current time step. After a possible proliferation event, the cell attempts to move, with probability $p_m$. If the cell performs a motility event, it changes position to one of the six neighboring lattice sites in its von Neumann neighborhood. In the case of unbiased migration all six sites are equally likely. However, we assume that the cells can sense their microenvironment consisting of white matter and blood vessels. The details of how the environmental cues influences cell migration is discussed

**Table 3**. **Clinical data for the four patient-derived cell lines used in this study.**

| Cell line | Tumor number | Subtype | Patient age | Patient sex | Survival (days) | Experiment duration (days) |
|---|---|---|---|---|---|---|
| U3013MG | 1, 2 | Proneural | 78 | Female | 122 | 60, 66 |
| U3230MG | 3, 4 | Mesenchymal | 61 | Male | 712 | 119, 95 |
| U3033MG | 5, 6 | Classical | 58 | Male | 178 | 165, 136 |
| U3062MG | 7, 8 | Mesenchymal | 76 | Female | 166 | 282, 275 |

in detail in the next section. The cell moves into the target site only if the site is below its carrying capacity. If the lattice site is full, the cell aborts its attempt to move. Once $N_t$ agents have been given the opportunity to proliferate and migrate, the simulation time is updated, $N_{t+1}$ computed, and the procedure is repeated for $T$ time steps.

As initial conditions we seed cells in a spherical region with radius of 8 grid points and center at the location corresponding to the injection site. In our 3D lattice this is the site located at $(94, 133, 115)$. In each voxel we place 3 cells, resulting in a total of 6327 cells. The boundary conditions are such that if a cell attempts to move out of the brain, the migration attempt is aborted and the cell remains in place.

**3.3.1 Influence of white matter and blood vasculature on migration.** In order to model the preferential motion of cells towards white matter and blood vessels, we introduce a bias to the random migration. We do that by assuming that each cell can perceive its surrounding region, and becomes more likely to move towards voxels containing white matter or blood vasculature. Let us consider the case of how cells are attracted towards white matter first. The aim is to compute how much the cell is influenced by white matter which is within a distance $d$ from the cell. This is done by first finding all voxels within a distance $d$ to the cell, which contains white matter. Let a particular cell be located at $\mathbf{x}_{cell}$. Assume that there are a total of $i$ voxels containing white matter, located at $\mathbf{x}_{wm}^{(i)}$, which our cell can sense. We then compute the vector pointing from $\mathbf{x}_{cell}$ towards $\mathbf{x}_{wm}^{(i)}$ and divide by the Euclidean distance:

$$\mathbf{v}^{(i)} = \frac{\mathbf{x}_{wm}^{(i)} - \mathbf{x}_{cell}}{||\mathbf{x}_{wm}^{(i)} - \mathbf{x}_{cell}||_2} = (x_i, y_i, z_i). \tag{1}$$

The vector $\mathbf{v}^{(i)}$ points in the direction of the white matter voxel, but has unit length. Recall that a migrating cell can move in either of the 6 directions in its von Neumann neighborhood. We therefore have to convert the direction vector $\mathbf{v}^{(i)}$ into a vector $\mathbf{u}^{(i)}$ of length 6 that represents how $\mathbf{v}^{(i)}$ influences the probability of migrating in each of the 6 directions. For instance, if $\mathbf{v}^{(i)} = (1, 0, 0)$ we want the migration to be towards the positive $x$–direction, whereas if $\mathbf{v}^{(i)} = (-1, 0, 0)$ we want the migration to be in the negative $x$–direction. We therefore define $\mathbf{u}^{(i)}$ have 6 elements corresponding to the positive and negative $x$–axis, the positive and negative $y$–axis, as well as the positive and negative $z$–axis. We then calculate

$$\mathbf{u}^i = \begin{pmatrix} \max(x_i, 0) \\ \min(x_i, 0) \\ \max(y_i, 0) \\ \min(y_i, 0) \\ \max(z_i, 0) \\ \min(z_i, 0) \end{pmatrix}.$$

Finally, we add up all the contributions from the $\mathbf{u}^{(i)}$'s and normalize to 1, so that we obtain vector containing probabilities to move in each of the 6 directions:

$$\mathbf{P}_{wm} = \frac{\sum_i \mathbf{u}^{(i)}}{||\sum_i \mathbf{u}^{(i)}||_1}$$

where $i$ is the index of number of neighboring voxels containing white matter. The exact same procedure is performed for the blood vasculature voxels, resulting in a vector of length 6, $\mathbf{P}_{bv} \in \mathbb{R}^6$. For both white matter and vasculature we use a sensing radius of $d = 5$ voxels. The interpretation should be that $\mathbf{P}_{wm}$ contains the probabilities to migrate in each of the 6 directions if the migration is entirely governed by white matter. However, different cell lines show different preferential movement towards different anatomical structures, so we introduce two key parameters representing the strength of attraction towards white matter, $w_{wm}$ and blood vessels, $w_{bv}$, respectively. Both parameters takes values between 0 and

1, and must satisfy $w_{wm} + w_{bv} \leq 1$. The probability to migrate in each of the 6 directions is given by

$$\mathbb{1}\frac{1}{6}(1 - w_{wm} - w_{bv}) + w_{wm}\mathbf{P}_{wm} + w_{bv}\mathbf{P}_{bv},$$

where $\mathbb{1}$ is a vector of ones, of length 6. The first term represents the isotropic part, and the second and third terms the contributions from white matter and vasculature respectively.

### 3.4 Parameter estimation using approximate Bayesian computation

In order to estimate the four model parameters $p_m$ (probability to migrate), $p_p$ (probability to proliferate), $w_{wm}$ (white matter preference) and $w_{bv}$ (blood vessel preference), we use a version of the Approximate Bayesian Computation method [37] (ABC). ABC methods are often used when the computation of a likelihood function is expensive or intractable. In short, we specify prior distributions for our four parameters which are assumed to be uniformly distributed, and then sample parameter values, perform a simulation and compare how well such a simulation matches the data at hand. We perform 5000 simulations, and choose to accept the top 1% best performing simulations. Although the underlying model is stochastic, we perform a single simulation per set of parameters. The justification for this is twofold. Firstly, there appears to be low variance between runs based on manual observations. Secondly, the parameter estimation procedure captures, among other sources of variability, the stochasticity of the model.

One of the major challenges with fitting model parameters to data is how to choose a summary statistic used to compare a simulated tumor with a real tumor. One of the most common metrics used for comparing spatial data is the Jaccard index, defined for sets $A$ and $B$ through

$$J(A, B) = \frac{|A \cap B|}{|A \cup B|},$$

which for binary images of tumors measures the amount of overlap relative to the total area from both tumors. However, such a measure performs poorly for certain morphologies, such as diffusely growing tumors with low cell density. Consider for example a checkerboard type of pattern in two images, one containing the white squares and one the black ones. Although by visual inspection two such patterns look the same, their Jaccard index will be 0. This is also true for metrics such as the $L^2$-norm applied to individual pixels. For this reason we have decided to use summary statistics based on geometric properties of the tumors, which are not based on spatial location of the tumors. For binary images we first find all connected components, using MATLABs built-in function **bwconncomp**, and then we compute the number of connected components, area, perimeter, filled area and eccentricity. These measures are obtained for each connected component in the binary image. Once calculated we define the summary statistics as

$$\mathbf{s} = \begin{pmatrix} \text{Num. connected components} \\ \text{Mean(Area)} \\ \text{std(Area)} \\ \text{mean(Eccentricity)} \\ \text{std(Eccentricity)} \\ \text{std(Perimeter)} \\ \text{max(Perimeter)} \\ \text{max(FilledArea)} \end{pmatrix}, \tag{2}$$

where "std" is the standard deviation. We denote the summary statistics of simulation $i$ by $\mathbf{s}_i$, and the one from a experimental tumor image by $\mathbf{s}$. In ABC it is common practice to define a threshold $\delta$ so that parameters satisfying

$$||\mathbf{s}_i - \mathbf{s}|| < \delta \tag{3}$$

are accepted, or by accepting a fixed proportion (1% or 0.1%) of simulations [38]. In this work we do the latter, and accept 1% of the 5000 simulations.

After having obtained a sample of 50 parameters of each type we perform local linear weighted regression [34] with an Epanechnikov kernel (the bandwidth is chosen to be the smallest value of $\delta$ that permits exactly 50 simulations) and a logit transform to ensure that the adjusted posterior samples lie within the same range as the respective prior [38]. Once an adjusted sample has been obtained we perform kernel smoothing using an Epanechnikov kernel with bandwidth 1/10, and use the mean value of the resulting estimated posterior density function as an estimate of the parameter value.

The parameters are randomly sampled from the uniform prior distributions

$$p_m \sim U(0, 1),$$
$$p_p \sim U(0, 0.015),$$
$$w_{wm} \sim U(0, 1),$$
$$w_{bv} \sim U(0, 1).$$

We choose a carrying capacity of $K = 3$ and perform 1800 time steps. At the end time we save the final simulation configuration, i.e. a 3D lattice where each voxel is populated with 0,1,2 or 3 cells. Although a slightly larger value would be more realistic, it would induce significant computational cost.

The mouse experiments run for different durations due to differences in tumor formation rate and growth rate, whereas each simulation consist of 1800 time steps. Because of this, a simulation time step corresponds to different durations in real time in the eight experiments. Once the model has been fitted to one image from each tumor, we rescale the fitted parameters to measure migration rate $P_m$ ($\mu m$/hour) and proliferation rate $P_p$ (1/day), i.e. the same units (per hour and per day) in all of the eight experiments. Prior to applying the method to experimental data we apply it to synthetic data to demonstrate that our summary statistics perform on par with the Jaccard index on synthetic datasets when no regression adjustment is applied, and considerably outperforms it after applying regression adjustment.

## Simulation study

In our simulation study to assess different similarity metrics, we ran 16 test cases with each of the four parameters at a high and a low value. Values used were $p_m \in \{0.2, 1\}, p_p \in \{0.0005, 0.005\}, w_{wm} \in \{0.1, 0.45\}, w_{bv} \in \{0.1, 0.45\}$. We thus obtained datasets, and the combination of parameter values are shown in Fig 3.

For each dataset we test three different methods to estimate the parameters. Each method result in a set of 50 accepted parameters. The three methods we compare are the Jaccard index and our geometric similarity measure with and without regression adjustment. When no regression adjustment is used, meaning that parameter estimates are chosen to be the mean of the accepted parameters, we refer to it as being a "direct" method.

For each of the 16 datasets we compute the relative error between true parameters $p_m, p_p, w_{wm}, w_{bv}$ and the estimated parameters $\hat{p}_m^{(i)}, \hat{p}_p^{(i)}, \hat{w}_{wm}^{(i)}, \hat{w}_{bv}^{(i)}$, for the 50 accepted parameters, $i = 1, 2, \ldots, 50$, using

$$E_i = \frac{1}{4} \left( \left| \frac{p_m - \hat{p}_m^{(i)}}{p_m} \right| + \left| \frac{p_p - \hat{p}_p^{(i)}}{p_p} \right| + \left| \frac{w_{wm} - \hat{w}_{wm}^{(i)}}{w_{wm}} \right| + \left| \frac{w_{bv} - \hat{w}_{bv}^{(i)}}{w_{bv}} \right| \right), \tag{4}$$

which is the absolute relative error, averaged over the four model parameters. The error of a given method is then given as the average error over the 50 accepted parameters:

$$E = \frac{1}{50} \sum_{i=1}^{50} E_i. \tag{5}$$

We find that the error $E$ defined in (5) averaged over the 16 datasets is 0.857 using the Jaccard measure, 0.856 when using the geometric measure, and 0.512 when using the geometric measure followed by regression adjustment. The errors for the 16 cases is shown in Fig 3.

Based on these results we decided to use the geometric measure with regression adjustment of similarity when fitting the model to the histological sections obtained from the PDC xenografts.

## 4 Discussion

We have proposed a novel model of glioblastoma growth in mice that explicitly incorporates both white matter and blood vasculature, in order to be able to capture a wider range of tumor morphologies. By fitting the model to eight images obtained from mouse experiments we find that the model replicates the overall tumor size and shape (diffusive vs. bulky) very well in most cases. Moreover, it is able to reproduce the growth along the corpus callosum that is a common feature of glioblastoma growth in mouse models and patients. Because we use a geometric measure of similarity rather than a similarity measure based on spatial overlap, like the Jaccard index, the exact positions of the simulated tumors are not always accurate (see e.g. tumor 3 in Fig 4), but instead we obtain a larger morphological similarity between real and simulated tumors.

The choice of similarity measure used to compare tumors is of utmost importance, and remains one of the greatest challenges when comparing tumor images in general. To some extent, the choice of metric will be influenced by the application at hand. In some cases the exact position of the tumor or its boundary would have higher priority, and in other applications the overall size and structure (bulky vs. diffuse, alignment with anatomical structures or not) will be of interest. Yet another consideration is whether the initial tumor location is known. If the model simulation is initialized at the incorrect site in the brain or if image misalignment is present, a measure based solely on spatial overlap will perform poorly. Based on tests on synthetic data (see Sect 2.2), we found that the geometric measure with regression adjustment performs better than the Jaccard index on average for our model, even in the case where the previously mentioned sources of error are not present.

Our investigation into the response to different hypothetical drug combinations demonstrated a considerable inter-tumor heterogeneity, where some tumors exhibited a clear trade-off while others were monotone in their response. Furthermore, these results highlight the utility of patient-specific models to elucidate the most fruitful treatment options. However, for such models to be clinically relevant they would need to be validated on much larger datasets.

Throughout the process of modelling the growth of tumor cells injected into mice, a number of sources of error and uncertainty are introduced. First, the DTI dataset and the blood vasculature are obtained from different mice, and hence needs to be aligned. Second, once a tumor has been simulated, a slice has to be chosen for comparison to the corresponding histology section. Currently this was done manually by visual inspection of the section and the white matter structures present. Third, when removing the needle during the tumor transplantation procedure, a small number of cells can follow the needle and initiate tumor growth at other locations than the injection site. Fourth, mice of different strains may not be anatomically identical.

When setting up a mathematical model, a number of simplifying assumptions have to be made. In our model the rules of motion of cells is such an example. We assumed that cells can sense and respond to presence of white matter and blood vasculature within a spherical region in its vicinity, and that the attraction is stronger the closer the cell is to the

region. The exact details of how cells in actuality respond to such cues is not fully known. However, in an agent-based model like ours it is a straightforward task to test other assumptions of motion. It is also possible to incorporate other mechanisms such as cell-cell adhesion, chemotaxis or subpopulations of cells with different behavior.

In this work we chose to develop an agent-based model, which is but one of many types of models that could be used. We made this choice due to their flexibility and ease with which one can implement different rules that dictate the cells' behavior and interactions with their environment. Another group of models are the continuum models in the form of partial differential equations describing the spatial and temporal evolution of the density of cells. These types of models can be phenomenologically motivated or derived from agent-based models. Agent-based models are best suited when the entity of interest is the cell, and where stochastic effects may play a role, whereas continuum models are more suitable for large populations of cells.

A number of important topics warrants further research. First, our model should be systematically benchmarked against other models to determine its performance, in particular after increasing the number of simulations used in the ABC method. Second, it should be determined whether the estimated parameters correlate with known markers of white matter invasion and perivascular invasion. To do that the model would have to be fitted to a larger number of tumors which are also characterised on the genomic and proteomic scale. Third, the model could be extended to include additional biological or physical mechanisms. For example explicit modelling of nutrient availability, and as a consequence, a proliferation rate which becomes dependent on proximity to vasculature. Finally, our model requires at least one observation of the spatial spread of a tumor in order to fit model parameters. The model can be used to characterize tumors as we have done in the current work, but the model can also be used to make future predictions. In this case it would be of great interest to investigate whether data obtained from next-generation sequencing technologies, such as RNA-seq and other high-throughput molecular sequencing methods, or *in vitro* experiments, could be integrated and used to estimate migration and proliferation.

## Supporting information

**S1 Fig. Images showing the prepossessing steps.**
(EPS)

## Author contributions

**Conceptualization:** Adam A. Malik, Philip Gerlee, Sven Nelander.

**Data curation:** Adam A. Malik, Cecilia Krona, Soumi Kundu.

**Formal analysis:** Adam A. Malik.

**Funding acquisition:** Philip Gerlee, Sven Nelander.

**Investigation:** Adam A. Malik, Cecilia Krona, Soumi Kundu, Philip Gerlee.

**Methodology:** Adam A. Malik, Cecilia Krona, Soumi Kundu, Philip Gerlee, Sven Nelander.

**Project administration:** Adam A. Malik, Sven Nelander.

**Resources:** Cecilia Krona, Soumi Kundu, Sven Nelander.

**Software:** Adam A. Malik.

**Supervision:** Philip Gerlee, Sven Nelander.

**Validation:** Adam A. Malik, Sven Nelander.

**Visualization:** Adam A. Malik.

**Writing – original draft:** Adam A. Malik, Cecilia Krona, Philip Gerlee, Sven Nelander.

**Writing – review & editing:** Adam A. Malik, Cecilia Krona, Philip Gerlee, Sven Nelander.

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
