## [Decision Letter · Decision Letter 0]

29 Apr 2025

PCOMPBIOL-D-24-02103

Anatomically aware simulation of patient-specific glioblastoma xenografts

PLOS Computational Biology

Dear Dr. Malik,

Thank you for submitting your manuscript to PLOS Computational Biology. After careful consideration, we feel that it has merit and are pleased to inform you that we are recommending a **minor revision**. While the manuscript does not fully meet PLOS Computational Biology's publication criteria as it currently stands, we invite you to submit a revised version of the manuscript that addresses the points raised during the review process.

Please submit your revised manuscript within 30 days. If you will need more time than this to complete your revisions, please reply to this message or contact the journal office at ploscompbiol@plos.org. Please include the following items when submitting your revised manuscript:

We look forward to receiving your revised manuscript.

Kind regards,

Heber L. Rocha, Ph.D.

Guest Editor

PLOS Computational Biology

Christoph Kaleta

Section Editor

PLOS Computational Biology

**Journal Requirements:**

At this stage, the following Authors/Authors require contributions: Adam Malik, Cecilia Krona, Soumi Kundu, Philip Gerlee, and Sven Nelander. Please ensure that the full contributions of each author are acknowledged in the "Add/Edit/Remove Authors" section of our submission form.

4) Your manuscript's sections are not in the correct order. Please amend to the following order: Abstract, Introduction, Results, Discussion, and Methods

5) Thank you for including an Ethics Statement for your study. Please include:

i) The full name(s) of the Institutional Review Board(s) or Ethics Committee(s).

6) Please upload all main figures as separate Figure files in .tif or .eps format. For more information about how to convert and format your figure files please see our guidelines:

7) Please ensure that all Figure files have corresponding citations and legends within the manuscript. Currently, Figure 2 does not have in-text citations. Please include the in-text citation of the figure. Please also ensure that Figure 2 is labeled.

8) We notice that your supplementary Figures, and information are included in the manuscript file. Please remove them and upload them with the file type 'Supporting Information'. Please ensure that each Supporting Information file has a legend listed in the manuscript after the references list. Please cite and label the supplementary figure as "S1 Figure".

9) Some material included in your submission may be copyrighted. According to PLOSu2019s copyright policy, authors who use figures or other material (e.g., graphics, clipart, maps) from another author or copyright holder must demonstrate or obtain permission to publish this material under the Creative Commons Attribution 4.0 International (CC BY 4.0) License used by PLOS journals. Please closely review the details of PLOSu2019s copyright requirements here: PLOS Licenses and Copyright. If you need to request permissions from a copyright holder, you may use PLOS's Copyright Content Permission form.

Potential Copyright Issues:

i) Figure 1. Please confirm whether you drew the images / clip-art within the figure panels by hand. If you did not draw the images, please provide (a) a link to the source of the images or icons and their license / terms of use; or (b) written permission from the copyright holder to publish the images or icons under our CC BY 4.0 license. Alternatively, you may replace the images with open source alternatives. See these open source resources you may use to replace images / clip-art:

10) Please amend your detailed Financial Disclosure statement. This is published with the article. It must therefore be completed in full sentences and contain the exact wording you wish to be published.

**Reviewers' comments:**

Reviewer's Responses to Questions

Reviewer #1: General summary: In this article the authors present a method to estimate tumor growth model parameters from histological sections from patient-derived xenografts. Using four different PDX models with 2 replicates each, the authors collected tissue sections which were subsequently binarized for comparison against the output of an agent based model. The agent base model described tumor growth and invasion which was sensitive to white matter fiber tracts (DTI collected in a different animal) and tissue-cleared maps of vasculature (Also collected in different animals). The authors then applied approximate bayesian computation (ABC) requiring sampling parameter ranges to yield 5000 simulations per animal, the top 1% (50) were selected using a geometry based measure. In general, this is a very interesting paper that presents a novel contribution to Computational Biology/Mathematical Oncology demonstrating an approach for comparing agent based model results to experimental data. I do have several concerns that should be addressed before further consideration.

Comment 1: Comparison to existing literature and selection of method: In the introduction you provide an overview of existing PDE (proliferation-invasion) and ABM based models. What is the main motivation for using ABM methods over PDE methods for these type of problems? This is covered in the discussion, but some of this topic should be covered in the introduction to help support the modeling choices made in this manuscript. Some PDE-based methods to consider discussing that include vasculature-coupled growth and/or white matter coupled growth:

(1) Human application: Hawkins-Daarud A, Rockne RC, Anderson AR, Swanson KR. Modeling Tumor-Associated Edema in Gliomas during Anti-Angiogenic Therapy and Its Impact on Imageable Tumor. Front Oncol. 2013 Apr 4;3:66. doi: 10.3389/fonc.2013.00066. PMID: 23577324; PMCID: PMC3616256.

(2) Mice/rats: Hormuth, D.A., Jarrett, A.M., Feng, X. et al. Calibrating a Predictive Model of Tumor Growth and Angiogenesis with Quantitative MRI. Ann Biomed Eng 47, 1539–1551 (2019). https://doi.org/10.1007/s10439-019-02262-9

(3) Mice/rats: Rutter, E.M., Stepien, T.L., Anderies, B.J. et al. Mathematical Analysis of Glioma Growth in a Murine Model. Sci Rep 7, 2508 (2017). https://doi.org/10.1038/s41598-017-02462-0

Comment 2: Third paragraph introduction: I think there is a word or phrase missing here. “This paper seeks to align two distinct approaches to brain tumor invasion”. Is it “two distinct approaches to model/study brain tumor invasion”?

Comment 3: Missing figure label and or caption? At the top of page 6, there are two panels showing the 3D maps of white matter and the vasculature. These should be given a figure number and referenced in the text.

Comment 4: Figure 3 is missing a figure caption.

Comment 5: Section 2.2, third paragraph. On the first read it is not clear why 50 accepted parameters were chosen. I would provide more background in this section to say how many total simulations/samples were collected and what percent of those you saved. (You might add this in your discussion of ABC in section 2.1.

Comment 6: Section 2.3, Some of the comparisons between replicates in this section are vague or contradictory to the results shown in Table 1. For example,for U3230MG you state that the parameters agree well except for w_bv, but the median or mean value of p_m is nearly 2 times different between the replicates. Similar comments for U3062MG.

Comment 7: Section 2.3, I would consider defining delta here as the acceptance threshold to improve the clarity of this section.

Comment 8: For figure 4, I would consider adding the error or the value of the Jaccard score to provide reference quantifications for the best fits.

Comment 9: Section 2.5. I am not entirely convinced that this section is needed for this manuscript. Is the proposed in silico drug mimicking a clinical available option? If not, is there a reason why standard of care treatment (e.g., fractionated radiotherapy and/or temozolomide) options were not modeled on with this framework

Comment 10: Section 3.1, “data” should be changed to “Data”.

Comment 11: Section 3.2. Second paragraph. What are the units of the radius of 8? (8 microns, lattice/grid positions?). How closely does this radius of 8 correspond to the volume of injected cells. Does the volume/radius of injected cells in the initial condition influence the estimated model parameters? Have you considered using an ellipsoid to model the initial condition (assuming cells are distributed along the injection track rather than at a precise location?

Comment 12: Section 3.3, “We choose a carrying capacity of K = 3”. How much does the selection of K influence the estimated parameters? Would a higher K (K = 5 or 10) or a lower K (K = 1 or 2) skew the ratio of the probability to proliferate and probability to migrate?

Comment 13: Discussion, Is only one tissue section available from the experimental data? If so, is a 3D simulation truly needed to estimate these parameters, and if not could the number of cells in the 2D simulation be increased?

Reviewer #2: Reviewer Summary:

This manuscript presents a promising computational approach that explicitly incorporates both white matter and blood vasculature to predict glioblastoma xenograft growth. The overall structure and summary of the work are well articulated. However, the manuscript would benefit from clarification and elaboration in several key areas.

Major Comments:

1) In the Results section, there is ambiguity in the use of the terms metric, method, and algorithm within the ABC framework. The manuscript should clarify that the same ABC rejection algorithm is used throughout, with a fixed tolerance (e.g., 1% acceptance rate). It is unclear what the authors mean by “regression adjustment”—does this refer to the use of the Epanechnikov kernel, or to a change in the summary statistics used to compare geometric metrics? Please revise the following paragraph accordingly, and ensure consistent terminology throughout the manuscript:

“Given such ‘ground truth’ data, we evaluated three versions of the ABC algorithm. First, in the simplest version, we matched simulations to experiments using the Jaccard index (the fraction of tumor-containing pixels that overlapped between experiment and simulation). We further proposed a geometric similarity measure with regression adjustment (as explained in Section 3.3), and a ‘direct’ method without regression adjustment.”

In addition, the explanation of regression adjustment in Section 3.3 is not sufficiently clear. More detail, possibly in the Supplementary Information (SI), would be helpful.

2) In Section 2.4, it is interesting that the full model demonstrated greater plausibility than the white-matter-only model for tumors 4, 5, and 7. Since these are the largest tumors, could this be due to their increased dependence on vasculature? A discussion of the biological rationale behind this finding would strengthen the manuscript.

3) If the time scale is adjusted for each experiment, how can the model be used to make predictions in scenarios where no prior data are available? Since the model uses a real-time unit, one could potentially estimate the growth pattern from other data sources (e.g., RNA-seq). Additionally, is the spatial unit also adjusted for each experiment? Please clarify.

4) The manuscript lacks information regarding model uncertainty. Given that this is a stochastic model, how many replicates per parameter set are required to produce an accurate summary statistic of the model output?

Minor Comments:

1) In the last paragraph of Section 2.2, please describe the meaning of “formula 4 and 5” to improve readability and flow.

2) Figures 2 and 3 lack captions and are not clearly referenced in the main text.

3) In Section 2.5, the authors state:

“For tumors 1–4 and 6, it appears that the cell count is lowest when the tumor is treated with only anti-migration or only anti-proliferation drugs, with a maximum somewhere between the two extreme cases.”

Does this observation not apply to tumor 5? Please clarify.

4) In SI Section 5.2.4, the comparison steps need more detail:

Step 1: What criterion was used for decision-making? Was it based on error?

Step 2: What technique was used in the binarization step?

Step 4: Please describe the “closest point” method in one sentence to give readers a quick understanding.

Step 5: Provide more detail about how the transformation was applied.

**Have the authors made all data and (if applicable) computational code underlying the findings in their manuscript fully available?**

Reviewer #1: Yes

Reviewer #2: Yes

PLOS authors have the option to publish the peer review history of their article (what does this mean?). If published, this will include your full peer review and any attached files.

Reviewer #1: No

Reviewer #2: No

**Figure resubmission:**
---

## [Decision Letter · Decision Letter 1]

5 Nov 2025

PCOMPBIOL-D-24-02103R1

Anatomically aware simulation of patient-specific glioblastoma xenografts

PLOS Computational Biology

Dear Dr. Malik,

Thank you for submitting your manuscript to PLOS Computational Biology. After careful consideration, we feel that it has merit but does not fully meet PLOS Computational Biology's publication criteria as it currently stands. Therefore, we invite you to submit a revised version of the manuscript that addresses the points raised during the review process. Please pay particular attention to the comments of reviewer 2, asking to integrate discussion points from the response letter into the manuscript.

We look forward to receiving your revised manuscript.

Kind regards,

Christoph Kaleta

Section Editor

PLOS Computational Biology

Christoph Kaleta

Section Editor

PLOS Computational Biology

**Journal Requirements:**

1) Please amend your detailed Financial Disclosure statement. This is published with the article. It must therefore be completed in full sentences and contain the exact wording you wish to be published.

2) Please ensure that the funders and grant numbers match between the Financial Disclosure field and the Funding Information tab in your submission form. Note that the funders must be provided in the same order in both places as well.

3) Please ensure that the files are uploaded in the online submission form in a correct numerical order

**Reviewers' comments:**

Reviewer's Responses to Questions

**Comments to the Authors:**

Reviewer #1: I have no further comments, the authors have addressed my concerns in in their revised manuscript.

Reviewer #2: The following points were addressed satisfactorily in the response but should still be integrated into the revised manuscript:

Q2) The authors provide a satisfactory conceptual response, but this point should be explicitly mentioned in the Discussion section as a potential model extension and limitation, particularly regarding proliferation dependency on vascular proximity and nutrient/oxygen availability.

Q3) The response is satisfactory and well reasoned, but the discussion should be expanded in the manuscript to reflect these limitations and potential extensions (e.g., need for prior data, possible integration of RNA-seq).

Q4) The justification is sound, but the manuscript should explicitly mention that one replicate per parameter set was used because of observed low variance, and that parameter estimation accounted for this low stochastic variability.

Minor comment:

Q3) The authors’ response indicates the description has been rewritten, but I cannot find this change in the revised manuscript. The clarification therefore appears incomplete. Please update the manuscript to include the full corrected explanation and indicate where this change was made.

**Have the authors made all data and (if applicable) computational code underlying the findings in their manuscript fully available?**

Reviewer #1: Yes

Reviewer #2: Yes

PLOS authors have the option to publish the peer review history of their article (what does this mean?). If published, this will include your full peer review and any attached files.

Reviewer #1: **Yes:** David A. Hormuth, II

Reviewer #2: No

**Figure resubmission:**
---

## [Decision Letter · Decision Letter 2]

10 Dec 2025

Dear Dr. Malik,

We are pleased to inform you that your manuscript 'Anatomically aware simulation of patient-specific glioblastoma xenografts' has been provisionally accepted for publication in PLOS Computational Biology.

Best regards,

Christoph Kaleta

Section Editor

PLOS Computational Biology

Christoph Kaleta

Section Editor

PLOS Computational Biology

Reviewer's Responses to Questions

**Comments to the Authors:**

Reviewer #2: Given that the authors have successfully incorporated the necessary changes into their revised manuscript, my concerns are fully resolved.

**Have the authors made all data and (if applicable) computational code underlying the findings in their manuscript fully available?**

Reviewer #2: Yes

PLOS authors have the option to publish the peer review history of their article (what does this mean?). If published, this will include your full peer review and any attached files.

Reviewer #2: No

---

## [Editor Report · Acceptance letter]

PCOMPBIOL-D-24-02103R2

Anatomically aware simulation of patient-specific glioblastoma xenografts

Dear Dr Malik,

I am pleased to inform you that your manuscript has been formally accepted for publication in PLOS Computational Biology. Your manuscript is now with our production department and you will be notified of the publication date in due course.

With kind regards,

Anita Estes
